# Radiomics and Clinical Biomarkers Improve Prediction of Vertebral Compression Fracture in Multiple Myeloma

**Djennifer Madzia-Madzou**[1] (ORCID)          D.K.MADZIA-MADZOU-3@UMCUTRECHT.NL
**Margot Jak**[2]                                           M.JAK@UMCUTRECHT.NL
**Bart de Keizer**[3]                                 B.DEKEIZER@UMCUTRECHT.NL
**Jorrit-Jan Verlaan**[4]                            J.J.VERLAAN@UMCUTRECHT.NL
**Nicole Erler**[5]                                     N.S.ERLER@UMCUTRECHT.NL
**Monique Minnema**[2]                             M.C.MINNEMA@UMCUTRECHT.NL
**Kenneth Gilhuijs**[1]                           K.G.A.GILHUIJS@UMCUTRECHT.NL

[1] *Imaging Sciences Institute,* [2] *Hematology,* [3] *Radiology,* [4] *Radiotherapy,* [5] *Data Science & Biostatistics, University Medical Center Utrecht, Utrecht, the Netherlands*

## Abstract

Vertebral compression fractures (VCFs) are a debilitating complication of multiple myeloma (MM), affecting up to 60% of patients. The Spine Instability Neoplastic Score (SINS), developed for solid tumour metastases, is commonly used but unvalidated for MM. We evaluated whether CT radiomics features improve VCF prediction compared to SINS. In a retrospective cohort of 271 MM patients (1,331 CT scans), 1,271 radiomics features were extracted and reduced to 121 robust features. Five Cox models were evaluated using internal 5-fold cross-validation with patient-level bootstraps (R=1,000). The full combined model (SINS + clinical biomarkers + radiomics) significantly outperformed SINS (C-index: 0.70 vs 0.57, $p = 0.003$; 2-year AUC: 0.71 vs 0.62, $p = 0.007$). Radiomics alone also showed significant improvement (C-index: 0.66, $p = 0.037$). CT radiomics and clinical biomarkers substantially improved VCF prediction over SINS, enabling personalized risk assessment for MM patients.

**Keywords:** Multiple myeloma, vertebral compression fracture, radiomics, SINS, survival analysis, multimodal, risk prediction

## 1. Introduction

Multiple myeloma (MM) is a hematological malignancy caused by the accumulation of abnormal plasma cells in the bone marrow. One of the severe complications of this patient group is vertebral compression fractures (VCFs) (Thorsteinsdottir et al., 2020). VCFs lead to pain managed by opioids, reduced lung capacity, limited mobility and an increase in mortality. As MM patients live longer with advances in treatment (Abdallah et al., 2023), protecting their spines becomes increasingly imperative.

The Spinal Instability Neoplastic Score (SINS) was developed as a referral tool for spinal instability caused by metastatic solid tumors (Fisher et al., 2010). Although the SINS has not been validated for MM patients, it is currently used clinically to refer patients with high-risk vertebrae to the orthopaedic surgeon. Given that up to 60% of MM patients develop VCFs after diagnosis (Zijlstra et al., 2023), a reliable screening tool is critically needed.

Radiomics offers the potential to capture vertebral phenotype beyond clinical scoring systems by extracting quantitative features from scans. The purpose of this study is to evaluate whether CT radiomics features and clinical biomarkers improve VCF prediction compared to SINS.

## 2. Material and Methods

### 2.1. Data

This retrospective study included a longitudinal cohort of 271 MM patients (median age 60 years [range, 34–82], 37% female, median 4 scans/patient) with two or more serial CT scans (1,331 total) from the University Medical Center Utrecht. IBSI-compliant radiomics features (1,271 total per vertebra) were extracted from the CT scans (Zwanenburg et al., 2020) and corresponding vertebral segmentations (Madzia-Madzou et al., 2025). A total of 33 clinical biomarkers were collected, including laboratory parameters, prior anti-myeloma treatment, and concomitant medication.

The primary outcome, a VCF, was defined as: (1) new fracture with $\geq 25\%$ height loss, (2) progression of existing collapse, or (3) surgical intervention treating vertebral collapse.

### 2.2. Feature Selection

Feature selection reduced radiomics features to 121 (prioritising robustness to resolution/noise and low inter-feature correlation). A Lasso-Cox regression model was used to first identify key predictors from the 154 radiomics and clinical parameters using a nested 5-fold cross-validation with 50 repeated cross-validations in the inner loop. The key predictors that were selected in at least 80% of the folds were: hemoglobin, vertebral fracture count, original shape MajorAxisLength, log sigma 5 mm first order 90Percentile, original shape LeastAxisLength and wavelet HLL first order Energy.

### 2.3. Models

Five Cox models were developed: (1) SINS model (including location, lesion quality, spinal alignment, posterolateral involvement, and vertebral body collapse); (2) clinical model (hemoglobin and prior fracture count); (3) radiomics model (original shape MajorAxisLength, log sigma 5 mm first order 90Percentile, original shape LeastAxisLength, and wavelet HLL first order Energy); (4) clinical + radiomics model; and (5) full combined model (all aforementioned parameters plus SINS).

### 2.4. Statistical Analysis

Internal 5-fold cross-validation with patient-level bootstrap (R=1,000) for 95% confidence intervals (CI). Metrics: C-index, 2-year risk Area Under the Curve (AUC), 2-year Integrated Calibration Index (ICI) and 2-year Brier score.

## 3. Results

The full combined model significantly outperformed SINS in discrimination (C-index: 0.70 [95% CI: 0.66-0.76] vs 0.57 [95% CI: 0.54-0.58], $p = 0.003$; 2-year AUC: 0.71 [95% CI:

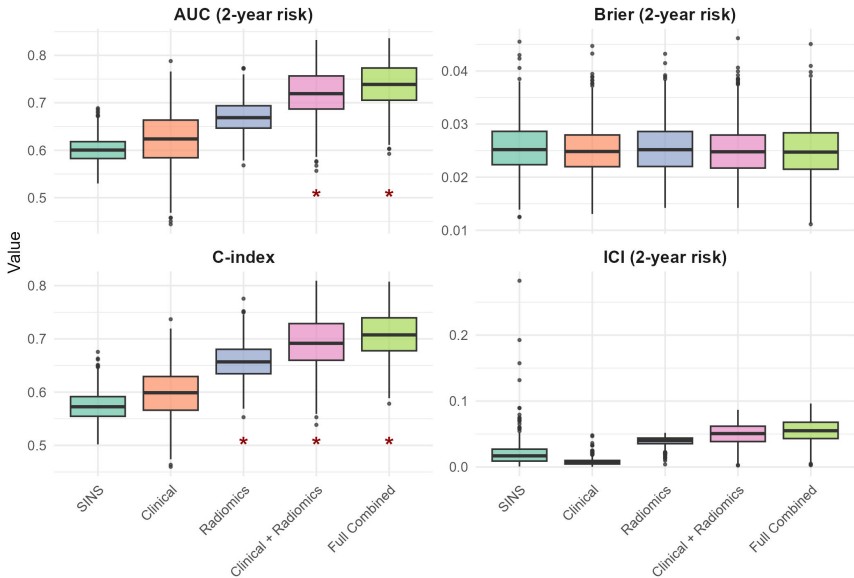

Figure 1: Bootstrap distributions (1,000 iterations) of C-index, 2-year Area Under the Curve (AUC) (higher is better), 2-year Integrated Calibration Index (ICI) and 2-year Brier score (lower is better). * indicates $p < 0.05$ compared to SINS.

0.64-0.80] vs 0.62 [95% CI: 0.58-0.63], $p = 0.007$). Radiomics alone also showed significant improvement in C-index (0.66 [95% CI: 0.62-0.72], $p = 0.037$). No significant differences were observed in ICI or Brier scores, indicating comparable calibration and overall accuracy (Figure 1).

## 4. Discussion

Our results demonstrate that CT radiomics and clinical biomarkers significantly improved VCF prediction over SINS in MM patients. The full combined model achieved a C-index of 0.70, representing a clinically meaningful improvement over SINS alone (0.57).

Several limitations warrant consideration. First, this was a single-center retrospective study; external validation is needed. Second, the mechanical pain component of SINS could not be assessed due to documentation limitations. Third, the sample size resulted in wide confidence intervals for some metrics.

In conclusion, CT radiomics substantially improved VCF prediction over SINS, enabling personalized risk assessment for MM patients.

## Acknowledgments

This work was supported by the Hanarth Fonds. We thank the data managers and clinical students who assisted with data collection.

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
