# OpenReview forum: "Radiomics and Clinical Biomarkers Improve Prediction of Vertebral Compression Fracture in Multiple Myeloma"
_MIDL.io/2026/Short_Papers — MIDL 2026 - Short Papers Poster_

### Official Review · Reviewer_3Gjc · 2026-05-07
**Radiomics and Clinical Biomarkers**

**Rating:** 5
**Confidence:** 5

**Review:**

The paper addresses a clinically important and understudied problem: prediction of vertebral compression fractures (VCFs) in multiple myeloma (MM) patients. The motivation is well justified because the currently used Spine Instability Neoplastic Score (SINS) was designed for metastatic solid tumors and lacks validation in MM. The authors propose a multimodal prognostic framework combining radiomics, clinical biomarkers, and SINS-derived variables within a survival modeling setting. The paper presents promising and clinically meaningful results, and the methodological direction is sound. While the current study is somewhat preliminary and would benefit from additional methodological transparency and external validation, it makes a valuable contribution to the emerging intersection of radiomics and risk prediction in hematologic oncology.

**Summary:**

This paper presents a multimodal survival prediction framework combining CT radiomics, clinical biomarkers, and SINS parameters to predict vertebral compression fractures (VCFs) in multiple myeloma patients. Using a retrospective longitudinal cohort and Cox-based modeling, the authors show that radiomics and clinical biomarkers significantly improve predictive discrimination over SINS alone. The study addresses an important clinical problem and demonstrates promising performance gains, although external validation and methodological clarifications are still needed.

**Strengths:**

Addresses an important and clinically relevant problem: prediction of vertebral compression fractures (VCFs) in multiple myeloma patients.
Uses a multimodal framework combining radiomics, clinical biomarkers, and SINS features.
Relatively large longitudinal dataset for this application
Appropriate use of survival analysis (Cox models) for time-to-fracture prediction.
Includes nested cross-validation and patient-level bootstrapping, which strengthens evaluation methodology.

**Weaknesses:**

Insufficient detail on handling repeated longitudinal scans, raising potential concerns about data leakage between folds.
No external validation cohort, limiting conclusions about generalizability.
Retrospective single-center design may reduce robustness across institutions and scanners.
High-dimensional radiomics setting with relatively limited sample size may still risk overfitting.
Limited biological or clinical interpretation of the selected radiomics features.

**Justification Of Rating:**

The paper presents a clinically relevant and methodologically solid study demonstrating that radiomics and clinical biomarkers improve vertebral compression fracture prediction over SINS in multiple myeloma patients. The problem is important, the multimodal approach is well motivated, and the reported gains in predictive performance are meaningful. The study also follows several good methodological practices, including IBSI-compliant radiomics extraction, nested cross-validation, and survival modeling.

---

### Decision · Program_Chairs · 2026-05-08

Accept (Poster)